# Clinical evaluation of the alcohol use disorders identification test (AUDIT) in Moshi, Tanzania

Joao Ricardo Nickenig Vissoci[1,2,3], Kaitlyn Friedman[1], Nayara Malheiros Caruzzo[4], Leonardo Pestillo de Oliveira[5], Alena Pauley[1,3], Siddhesh Zadey[2,3], Vanessa Menegassi[4], Francis Sakita[6], Judith Boshe[6], Catherine A. Staton[1,2,3]*, Blandina Mmbaga[1,6,7,8]

**1** Duke Global Health Institute, Duke University, Durham, NC, United States of America, **2** Duke Emergency Medicine, Duke University Medical Center, Durham, NC, United States of America, **3** Global Emergency Medicine Innovation and Implementation (GEMINI) Research Center, Duke University Durham, NC, United States of America, **4** State University of Maringá, Maringá, Paraná, Brazil, **5** Cesumar Institute for Science, Technology and Innovation, Unicesumar, Maringá, Paraná, Brazil, **6** Kilimanjaro Christian Medical Centre, Moshi, Tanzania, **7** Kilimanjaro Clinical Research Institute, Moshi, Tanzania, **8** Kilimanjaro Christian Medical University College, Moshi, Tanzania

* catherine.staton@duke.edu

**Data Availability Statement:** Data are only available upon request as data transfer requires a written agreement approved by the National Institute for Medical Research (Tanzania). Data

## Abstract

### Background

Alcohol use disorder is a major cause of morbidity and mortality in low- and middle-income countries. Alcohol screening using a validated tool is a useful way to capture high-risk patients and engage them in early harm reduction interventions. Our objectives were to 1) evaluate the psychometric evidence the Alcohol Use Disorders Identification Test (AUDIT) and its subscales in the general population of Moshi, Tanzania, and 2) evaluate the usefulness of the tool at predicting alcohol-related harms.

### Methods

Two hundred and fifty-nine adults living in Moshi, Tanzania were included in the study. We used the AUDIT and its subscales to determine the classification of harmful and hazardous drinking. To analyze the internal structure of AUDIT and the model adequacy we used Confirmatory Factor Analysis (CFA). The reliability of AUDIT was analyzed for Cronbach's alpha, Omega 6 and Composite Reliability. The optimal cut off point for the AUDIT was determined by the receiver operating characteristic (ROC) curve, using the Youden approach to maximize sensitivity and specificity.

### Results

The median score of the AUDIT was 1 (inter-quartile range: 0–7). The internal structure of the AUDIT showed factor loadings ranging from 0.420 to 0.873. Cronbach's alpha, Omega and Composite Reliability produced values above 0.70. The Average Variance Extracted

inquiries can be sent to Gwamaka W. Nselela at gwamakawilliam14@gmail.com.

**Funding:** The funders had no role in study design, data collection and analysis, decision to publish, or preparation of the manuscript. This project was supported by the Fogarty International Center of the National Institutes of Health (https://www.fic.nih.gov/) under Award Number K01TW010000 (PI, CAS).

**Competing interests:** The authors have declared that no competing interests exist.

was 0.530. For the AUDIT, a score of 8 was identified as the ideal cut-off value in our population.

## Conclusions

This study validates AUDIT in the general population of Moshi and is one of the only studies in Africa to include measures of the internal structure of the AUDIT and its subscales.

## 1 Introduction

Alcohol use disorders (AUD) are a major cause of global morbidity and mortality that disproportionately impacts low- and middle-income countries (LMICs) [1–3]. Alcohol use contributes 2–6% of the overall attributable risk of death, mostly due to injury [4–6]. This burden is especially large in the WHO Africa Region, which reports the highest rates of alcohol-attributable deaths and alcohol-attributable deaths caused by unintentional injuries worldwide [7]. In the last 20 years, alcohol-related deaths have increased by over 40% in Eastern sub-Saharan Africa [8, 9]. However, alcohol use is often underreported in surveys contributing to a likely underestimation of the severity of impact [10, 11].

There are numerous validated ways to screen individuals for unhealthy alcohol use and diagnose alcohol use disorders (AUD), including the Diagnostic and Statistical Manual of Mental Disorders (DSM), Alcohol Use Disorders Identification Test (AUDIT) and the subscales of the AUDIT. Previous versions of the DSM identified categories of "alcohol abuse," or "alcohol dependence," but the current DSM-5 uses 11 criteria to evaluate the unidimensional construct "alcohol use disorder" and categorizes individuals into three levels (mild, moderate, or severe AUD). The AUDIT, developed by the World Health Organization (WHO) in 1993, is a 10-item questionnaire that identifies harmful or hazardous drinking and alcohol dependence [12]. The AUDIT is a screening tool (rather than a diagnostic tool) intended for use in primary care settings. The AUDIT is typically used in a one-dimensional model, where a single total score helps to identify individuals with unhealthy alcohol consumption who are in need of further diagnostic follow-up and potential treatment [13].

Consistent alcohol screening using validated tools such as AUDIT is useful in capturing high-risk patients and engaging them in early harm reduction interventions [14]. Despite being a 10-item instrument, AUDIT is considered long for busy clinical environments around the world, which has led to the development of the AUDIT subscales [15–17]. These subscales, which include AUDIT-C, AUDIT-3, AUDIT-4, and AUDIT-PC, still screen for unhealthy alcohol use but contain only 1 to 5 items from the full AUDIT-10. In summary as seen in **Table 1**, AUDIT-Consumption (AUDIT-C) contains 3 questions about alcohol consumption [18, 19]. AUDIT-3 consists of a single question (3rd question of the full AUDIT) about binge

**Table 1. Subscales of the AUDIT versions.**

| Version | Items | Measure | Score |
|---------|-------|---------|-------|
| AUDIT-3 | Item 3 | Binge drinking [20] | Score ranges from 0–4 |
| AUDIT-C | Items 1, 2, 3 | Alcohol consumption [18] | Score ranges from 0–12 |
| AUDIT-4 | Items 1, 2, 3 and 10 | Alcohol consumption [21] | Score ranges from 0–16 |
| AUDIT-5 | Items 2, 4, 5, 9 and 10 | Problem drinking, alcohol dependence [22] | Scores ranges from 0–20 |
| AUDIT-PC | Items 1, 2, 4, 5 and 10 | Hazardous alcohol intake [23] | Scores ranges from 0–20 |

drinking [20], AUDIT-4 includes all the questions AUDIT-C with the addition of the 10th item from AUDIT-10 [21]. AUDIT-5 screens for problematic drinking and alcohol dependence [22], and AUDIT-(Piccinelli) Consumption (AUDIT-PC) is designed to screen for hazardous alcohol intake [23].

For comprehensive healthcare and management, it is important to validate the full AUDIT and its subscales in different settings and global populations to best screen for harmful and hazardous alcohol use. Since its development, the AUDIT has been validated in numerous high-, middle- and low-income countries including Germany [24], Brazil [25], Nepal [26], and India [27]. Even though one of the original development sites is in Kenya, there has been little subsequent work to validate the complete psychometric properties of the tool in the African continent or in the Swahili language. Previous work throughout Africa has included both reliability measures (Cronbach's alpha ranging from 0.83–0.98) [28–31], and preliminary validation as compared to the International Classification of Diseases 10 (ICD-10) or the Mini International Neuropsychiatric Interview questionnaire (MINI) for determining the criteria of alcohol use (areas under the curve ranging from 0.75–0.98) [28, 30, 32, 33]. However, there has been limited evaluation of the scale's internal structure. Our group recently conducted the first formal validation of the AUDIT in Tanzanian Swahili which included analysis of the internal structure, but this has not yet been replicated in the general population and is valid specifically for the traumatic brain injury (TBI) patients [28]. Hence, AUDIT has not yet been formally validated in the general population in this region, thus limiting the external validity of the tool.

We aimed to improve upon previous research by assessing the reliability, validity, and internal structure of the full AUDIT and its 5 subscales as a screening instrument in a sample drawn from the general population of Moshi, Tanzania. To ensure instrument validity, there must be substantial supporting evidence based on: (a) content of questions included in the instrument, (b) response process, (c) internal structure of the instrument, and (d) relationship to other variables/instruments [34].

Our objectives were to 1) evaluate the psychometric evidence of the AUDIT and its subscales in the general population (content, response, internal structure) and to test the unidimensional hypothesis of the instruments, and 2) evaluate AUDIT's usefulness in predicting alcohol-related harms. These aims will allow for a better assessment of the utility of the AUDIT and its subscales to screen for harmful and hazardous alcohol use in the general population of Moshi, Tanzania.

## 2 Materials & methods

### 2.1 Ethics

The study was approved by the Institutional Review Board of Duke University (IRB #Pro000061652), the Ethics Committee of the Kilimanjaro Christian Medical Center, Moshi, Tanzania and the National Institute of Medical Research in Tanzania. All participants provided written informed consent authorizing the collection and use of the data in this research.

### 2.2 Study setting

Moshi is a city in the Kilimanjaro region of Northern Tanzania that is home to the Kilimanjaro Christian Medical Centre (KCMC). KCMC is the third largest hospital in the country [35, 36] serving both the urban and rural population of Moshi, and is also a referral center for northwestern Tanzania. Therefore, KCMC was selected as a central location to assess general patterns of alcohol use in this region. Prior literature shows that the Moshi population has high proportions of alcohol use, specifically in youth [37] and in bar workers [38]. The burden of alcohol use in Moshi is almost twice that of the surrounding WHO Africa region [7]. In the

Moshi community, alcohol use is prevalent in 7.0% of women with partners, 9.3% in women without partners, and 22.8% in men [39], whereas it is prevalent in only 3.7% of the WHO Africa region population [7].

## 2.3 Study participants

A cross-sectional study was conducted and participants included 259 adults selected from different parts of the Moshi Urban community, at convenience. Our research assistants approached individuals on the streets, inviting them to participate in the study. Individuals were included in the study if they agreed to answer the questions, were at least 18 years of age, spoke English or Swahili, and provided informed consent. Participants were excluded if they reported not drinking alcohol in their lifetime. Data was collected by research assistants fluent in both Swahili and English, with experience in collecting research data using the AUDIT questionnaire [12] and trained previously by the researchers (CS, JV) in collecting data using these questionnaires and screening questions.

## 2.4 Instruments

We used an 11-question survey tool based on the DSM-5 as a gold standard to diagnose an AUD [40]. While DSM-5 validity remains to be studied, it remains the strongest gold standard self-reporting tool for the region; previously, another version, DSM-IV, has been validated in Northern Tanzania [41], yet its limited specificity and low sensitivity suggest DSM-5 would be a better scale. An AUD diagnosis using DSM-5 requires at least 2 of the 11 criteria in the past 12 months that assess alcohol use and alcohol dependence. Based on these DSM-5 criteria, an AUD is classified as mild (2 or 3 criteria), moderate (4 or 5 criteria), and severe (6 or more criteria). Questions about alcohol consumption over the past twelve months were collected; non-drinkers were defined as those who had consumed no alcohol over the preceding year and these participants were excluded. In terms of quantifying the number of drinks consumed, as there is no accepted international standard, the National Institute on Alcohol Abuse and Alcoholism (NIAAA) recommended guidelines for standard drinks were used [42]. Local researchers were trained to recognize and explain the NIAAA guidelines for standard drinks for regulated and unregulated alcohol common in the region, and have used these guidelines in numerous prior projects.

   We used the AUDIT to determine the classification of harmful and hazardous drinking that would be compared with the results from the DSM-5 based survey tool. In the 10-item questionnaire, each item is scored on a five-point Likert scale (from 0 to 4) and the overall score ranges from 0 to 40. The AUDIT subscales use various items of the AUDIT (**Table 1**). The 10-item sum score, when used as a unidimensional scale, has been shown to perform better in this setting, given the high association between alcohol use and alcohol dependence [28]. The internationally accepted AUDIT score of 8 is widely used as a cut-off for intervention in clinical and research settings [43].

## 2.5 Translation and adaptation

All processes to translate the DSM-5-based survey tool were conducted in accordance with WHO guidelines for health outcomes translation [44]. First, the instrument was translated into Swahili by a native translator, then another bilingual translator translated the Swahili version back into English. Finally, the English version was compared with the original version by another independent translator who was responsible for verifying the inconsistencies. The translation, adaptation, and content validation were supervised by a committee of five researchers. This panel consisted of Tanzanian researchers with backgrounds in health

sciences, psychiatry, and clinical research, as well as two international researchers with experience in acute care, alcohol use, and psychometrics. The panel reviewed the translated questions and discussed adaptations to the content that would improve the meaning of the items in relation to Tanzanian culture and the Swahili language. Translations were conducted by independent Swahili/English-speaking research assistants and evaluated by the panel. Changes were made according to the panel's feedback until a consensus was reached on the content of each item.

This tool was piloted with a sample of 10 Tanzanian adults to verify the coherence of language and content of the instrument. Minor grammar, verbiage, and spelling changes were made to improve comprehension for all educational levels. The last Swahili version was analyzed by a group of bilingual Tanzanian research nurses who evaluated the practical relevance, language clarity of the translated instrument, and theoretical coherence of the item. This analysis was evaluated by a five-point Likert scale. Finally, a focus group among these research nurses was conducted to improve the quality of the translations and discuss any discordances. We only used the DSM-5 AUD screening questions as the basis for translation which did not require a permission.

The process of translation, adaptation, and content validation of the AUDIT in this setting has been previously published [28]. AUDIT is available in the public domain and hence does not require a license to use, translate, or adapt.

Both English and Swahili versions of the survey instruments have been made publicly available here by presenting them in Supporting Information.

## 2.6 Data collection

Participants were approached in the Moshi community and asked to participate in this study. Upon meeting inclusion criteria and providing informed consent, participants were enrolled in the study. All participants then completed a 45-minute interview in which the DSM-5-based survey tool and the AUDIT were administered.

All data was collected on paper forms by trained research nurses and then entered into a REDCap database [45]. Quality assessment was performed at three points: at the conclusion of the data collection by the research nurses conducting the study; at data entry by the data entry personnel; and when the study principal investigator reviewed the REDCap dataset.

## 2.7 Data analysis

All analyses in this study were conducted with R Language for Statistical Computing [46]. For the descriptive analysis of sociodemographic variables, data were presented with measures of central tendency (means or medians), dispersion (standard deviations or interquartile range), or absolute and relative frequencies.

**2.7.1 Measuring internal structure.** To analyze the internal structure of AUDIT and the unidimensional model adequacy we used Confirmatory Factor Analysis (CFA). To test the CFA model adequacy we used a Weighted Least Square Means and Variance Adjusted (WLSMV) estimation including, Chi-square ($X^2$ and p-value), Root Mean Square Error of Approximation (RMSEA <0.08, I.C. 90%), Tucker-Lewis index (TLI> 0.90), and Comparative Fit Index (CFI> 0.95). All indices were evaluated according to the reference literature [47]. We also calculated the Average Variance Extracted (AVE) considering values greater than 0.50 as acceptable indicators [48].

**2.7.2 Measuring reliability.** The reliability of AUDIT was analyzed for Cronbach's alpha. Values above 0.80 are considered acceptable by Nunnally and Bernstein [49]. The composite reliability (CR) and Omega 6 coefficient were calculated using results from the CFA. Average Variance Extracted (AVE) was used to measure the level of variance captured by the construct

versus the level due to measurement error, with values of AVE above 0.5 considered acceptable.

**2.7.3 Measuring validity.** Content validity was evaluated by a Content Validity Coefficient for each item (CVCi) and for the instruments in total (CVCt) [50]. AUDIT's ability to determine harmful or hazardous alcohol use was compared to the gold standard DSM-5-based survey tool by calculating sensitivity, specificity, and area under the curve. The optimal cut-off point for the AUDIT was determined by the receiver operating characteristic (ROC) curve, using the Youden approach [51] to maximize sensitivity and specificity with an additional analysis using the maximum sensitivity approach.

# 3 Results

## 3.1 Sample characteristics

Fifty-two percent of the participants were male (n = 134) and the average age was 43 years old (SD = 15.91). The median score of the AUDIT was 1 (interquartile range: 0–7) for this population. One hundred and thirty-two (51.3%) participants had AUD as per DSM criteria with 41 mild, 28moderate, and 63severe cases of AUD.

## 3.2 Internal structure

The internal structure of the AUDIT performed well, with all items showing factor loadings ranging from 0.420 to 0.873 in the study population (**Table 2**).

## 3.3 Evidence of reliability

The AUDIT also demonstrated internal consistency. Cronbach's alpha, Omega, and Composite Reliability were calculated and produced values above 0.80 (**Table 2**). The AUDIT also presented adequate fit indices indicators for unidimensional models (**Table 2**). The AVE for the study population was 0.530.

The AUDIT subscales also showed good values of reliability and adequate fit indices for the unidimensional model. Their reliability indices (Cronbach's Alpha) can be observed in **Table 3**.

**Table 2. Reliability and confirmatory factor analysis (CFA) model fit indicators of the AUDIT.**

| AUDIT | Study Population |
|---|---|
| Reliability | |
| Cronbach's Alpha | 0.84 |
| Omega 6 | 0.85 |
| Composite Reliability | 0.92 |
| CFA | |
| $X^2$ (Df) / P-value | 56.459 (34) / 0.009 |
| RMSEA | 0.051 |
| TLI | 0.98 |
| CFI | 0.98 |
| AVE | 0.530 |
| Factor loadings Range Min-Max | 0.420–0.873 |

Note: CFA = Confirmatory Factor Analysis; $X^2$ = Chi-Square;Df = Degree of Freedom; RMSEA = Root Mean Square Error of Approximation; TLI = Tucker-Lewis Index; CFI = Comparative Fit Index; AVE = Average Variance Extracted.

**Table 3. AUCs, cut-off scores and reliability indices of the AUDIT its subscales.**

| | Youden Index method | | | | Maximum Sensitivity (= 1.00) method | | | | |
|---|---|---|---|---|---|---|---|---|---|
| | AUC (95% CI) | Cut-off | Sensitivity/ Specificity | PPV/ NPV | AUC (95%CI) | Cut-off | Specificity | PPV/ NPV | Cronbach's Alpha |
| **AUDIT** | 0.860 (0.813–0.907) | 8 | 0.77/0.78 | 0.66/0.86 | 0.860 (0.813–0.907) | 1 | 0.00 | 0.35/NaN | 0.84 |
| **AUDIT-3** | 0.755 (0.696–0.813) | 1 | 0.64/0.83 | 0.67/0.81 | 0.755 (0.696–0.813) | 1 | 0.00 | 0.35/NaN | - |
| **AUDIT-4** | 0.824 (0.773–0.874) | 5 | 0.81/0.67 | 0.58/0.87 | 0.824 (0.773–0.874) | 1 | 0.00 | 0.35/NaN | 0.68 |
| **AUDIT-5** | 0.828 (0.776–0.881) | 5 | 0.66/0.85 | 0.71/0.82 | 0.828 (0.776–0.881) | 1 | 0.00 | 0.35/NaN | 0.70 |
| **AUDIT-C** | 0.762 (0.690–0.834) | 5 | 0.57/0.85 | 0.59/0.84 | 0.821 (0.770–0.873) | 1 | 0.00 | 0.35/NaN | 0.71 |
| **AUDIT-PC** | 0.830 (0.780–0.881) | 5 | 0.82/0.67 | 0.58/0.87 | 0.830 (0.780–0.881) | 1 | 0.00 | 0.35/NaN | 0.70 |

Note: AUDIT-3 consists of only 1 item, hence, there is no reliability analysis for this version.

### 3.4 Evidence of validity

We calculated the AUDIT cut-off based on ROC curves, to screen for harmful and hazardous alcohol use requiring clinical intervention. For the AUDIT, a score of 8 was identified as the ideal cut-off value in our population. The version that showed better sensitivity for the general population was the AUDIT-PC (0.82), while better specificity was found in the AUDIT-5 and AUDIT-C (0.85). All of the cut-off and sensitivity and specificity scores by both approaches can be observed in **Table 3**. Additionally, the sensitivity and specificity for all AUDIT scores ranging from 1 to 34 are presented in **S1 Table in S1 File**.

## 4 Discussion

This is the first study to evaluate the psychometric evidence of the AUDIT in the general (i.e. non-clinical) population of Moshi, Tanzania. The conclusions drawn from this study build upon and reinforce results from previous validation studies of AUDIT and its subscales in South Africa and Namibia, and suggest that the subscales may be an appropriate replacement for the full AUDIT in busy clinical settings [16, 17, 20]. Further research is needed on the effectiveness of AUDIT and the subscales among women and specialized populations within Moshi, as past literature has suggested that these tools do not accurately measure unhealthy alcohol use in these groups [52].

This study is also one of the only studies in the African continent to include measures of the internal structure of the AUDIT and its subscales. Additionally, this is the second study validating the complete psychometric profile of the AUDIT in this setting, further confirming the external validity of the screening tool throughout Tanzania beyond a TBI population [28]. Overall, we found that the AUDIT had excellent internal consistency, reliability, and validity in this population comparable to that found in studies conducted in other African countries [16, 53].

Using predictive modeling, we determined that our results are in accordance with international standards and published literature: a cut-off of 8 for the AUDIT denotes harmful or hazardous drinking that may need clinical intervention. Despite demonstrating excellent validity with a cut-off score of 8, we are unable to draw conclusions concerning cut-offs for higher risk alcohol use in this population due to the lower specificity demonstrated in the ROC curves. The specific sample of alcohol users recruited in the current study along with possible limited statistical power also prevents us from specifying cut-offs for high-risk alcohol use. Previous validation attempts in diverse settings have also come across similar barriers with this delineation [30, 54, 55]. Yet, this is not likely a pertinent limitation in our setting; in Tanzania, there are limited interventions specifically designed for higher-risk drinking (e.g. AUDIT scores of

over 18), so upper limits of risky drinking are less pertinent as all levels of risk patients are offered the same resources.

This study had some limitations which need to be taken into account. This evaluation was performed in the language of Swahili, which is commonly spoken in Moshi in addition to tribal languages. It is possible that the tool would behave differently if delivered in tribal languages. However, there are over 100 tribal languages spoken in the country and Swahili is the most commonly used language. The study is also limited by the use of the AUDIT. The sensitivity and specificity of the overall AUDIT cut-off score of 8 can lead to some people with AUDs being missed while others without AUD presentation being pursued for further screening. The AUDIT does not perform well in identifying people with AUD, women in particular, and has been found to give a higher rate of false positives in countries with a low AUD prevalence compared to countries with a high AUD prevalence [56]. While different forms of screening for AUDs should be explored in future research, the AUDIT has still been widely used, validated, and is frequently reported to identify people with unhealthy alcohol habits. This study used DSM-5 which is considered a gold standard, for AUD clinical diagnosis in the United States and several other English-speaking parts of the world. However, validity of DSM-5 itself still needs to be studied in Tanzania; that said, this is our best available gold standard tool. Another point that must be taken into consideration is that in some cultures, there is a common practice of drinking from non-standard containers as well as sharing drinks with a group of people, which can represent a barrier to getting a reliable measure of alcohol consumption [20]. In order to reduce the impact of this limitation, we created culturally relevant estimates based on traditional alcohol percentages and drinking container amounts to estimate consumption. It is also important to highlight that the study, while drawn from the general population of Moshi (Northern Tanzania), was not representative of the population. Further, these findings may not be transferable to other communities or other parts of Tanzania with differing sociodemographic characteristics. Therefore, further research with diverse and representative samples is needed.

## 5 Conclusions

The collection of our work in Moshi, Tanzania has strengthened the power of the AUDIT and its subscales and suggests that it may be employed throughout the Kilimanjaro region in future public health efforts. Future research may focus on creating innovative interventions to reduce harmful or hazardous alcohol use disorders.

## Supporting information

**S1 File. AUDIT screening parameters.** Study Information & Consent Form (English): Study Information & Consent Form (Swahili): AUD Screening questions–DSM-V (English): AUDIT (English):AUD Screening questions–DSM-V (Swahili): AUDIT (Swahili).
(DOCX)

## Acknowledgments

We would like to acknowledge the KCMC/Duke Collaboration and the Emergency Medicine research team without whom none of this research would be possible.

## Author Contributions

**Conceptualization:** Joao Ricardo Nickenig Vissoci, Catherine A. Staton, Blandina Mmbaga.

**Data curation:** Nayara Malheiros Caruzzo.

**Formal analysis:** Joao Ricardo Nickenig Vissoci, Nayara Malheiros Caruzzo, Siddhesh Zadey, Vanessa Menegassi.

**Funding acquisition:** Catherine A. Staton.

**Investigation:** Francis Sakita, Judith Boshe.

**Methodology:** Joao Ricardo Nickenig Vissoci, Leonardo Pestillo de Oliveira.

**Project administration:** Catherine A. Staton.

**Software:** Joao Ricardo Nickenig Vissoci, Leonardo Pestillo de Oliveira, Siddhesh Zadey, Vanessa Menegassi.

**Supervision:** Catherine A. Staton, Blandina Mmbaga.

**Validation:** Leonardo Pestillo de Oliveira.

**Writing – original draft:** Kaitlyn Friedman, Nayara Malheiros Caruzzo, Catherine A. Staton.

**Writing – review & editing:** Joao Ricardo Nickenig Vissoci, Kaitlyn Friedman, Alena Pauley, Siddhesh Zadey, Vanessa Menegassi, Francis Sakita, Judith Boshe, Catherine A. Staton, Blandina Mmbaga.

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
