## [Decision Letter · Decision Letter 0]

22 Sep 2022

PONE-D-22-15181Clinical validation of the Alcohol Use Disorders Identification Test (AUDIT) in Moshi, TanzaniaPLOS ONE

Dear Dr. Catherine A. Staton,

Thank you for submitting your manuscript to PLOS ONE. After careful consideration, we feel that it has merit but does not fully meet PLOS ONE’s publication criteria as it currently stands. Therefore, we invite you to submit a revised version of the manuscript that addresses the points raised during the review process.

We look forward to receiving your revised manuscript.

Kind regards,

Sebsibe Tadesse, PhD

Academic Editor

PLOS ONE

Journal Requirements:

Reviewers' comments:

Reviewer's Responses to Questions

**Comments to the Author**

1. Is the manuscript technically sound, and do the data support the conclusions?

Reviewer #1: Yes

Reviewer #2: No

2. Has the statistical analysis been performed appropriately and rigorously? 

Reviewer #1: Yes

Reviewer #2: Yes

3. Have the authors made all data underlying the findings in their manuscript fully available?

Reviewer #1: No

Reviewer #2: No

4. Is the manuscript presented in an intelligible fashion and written in standard English?

Reviewer #1: Yes

Reviewer #2: Yes

5. Review Comments to the Author

Reviewer #1: I thank you for the opportunity to read the article which deals with a very important area within alcohol research, namely validating measuring instruments in different cultures. The aim of this study was to validate the AUDIT and its subscales in Tanzania and also to investigate the predictive value of these. However I identified some deficiencies.

The sample was an availability sample of 259 people who were asked on the street to complete questionnaires, which means that it is difficult to generalize the results to others than those who were not asked.

On page 13, the authors define reliability and validity. The definition of validity is limited to conceptual validity. Reliability can also be defined as the relationship between different items in the scale.

To diagnose alcohol use disorder, a self-rating scale with 11 items based on DSM-5 was used. However, there is no information on the validity of the scale. To quantify standard drinks, the American norm for standard drink has been used. However, it may be difficult for the participants in the study to translate the amount of alcohol they drank into standard American drinks.

The data has been analyzed with confirmatory factor analysis, which is a hypothesis testing method, but there is no information about which hypothesis is being tested. It should be clearer that it is a hypothesis about a one-dimensional model that is tested with a confirmatory factor analysis. Convergent validity means a relationship between two measuring instruments, but the authors have not stated which instrument was used in the calculation. The authors write that 127 participants had AUD divided into 50 mild, 30 moderate and 72 severe. Then the total will be 152 participants.

In terms of sensitivity and specificity, the numbers in the text does not agree with what is stated in table 3.

Reviewer #2: The study entitled "Clinical validation of the Alcohol Use Disorders Identification Test (AUDIT) in Moshi, Tanzania aim to validate the Alcohol Use Disorders Identification Test (AUDIT) in the general population of Moshi (Tanzania), and evaluate the usefulness of the tool at predicting alcohol-related harms.

The topic is interesting given the importance of alcohol-related health problems and the study is well conducted. However, there are important limitations that seems difficult to solve. Basically, the study is more a translation and adaptation than a validation. This is particularly so, because a convenience sample of 259 adults from Moshi (Northern Tanzania) con not be generalise to the whole country. The authors themselves correctly acknowledge this important limitation: “[the sample ]was not representative of the population”

The following commentaries should also be considered in order to improve the quality of the article for publication. I would like to emphasize that the study is very well conducted but with a very limited and not representative sample.

Methods

- Excluding participants who do not drink alcohol is not a good idea because prevent the study to examine prevalence of alcohol and risk overestimating alcohol related disorders. They should have been included and categorised with 0.

Data analysis and results.

- It is not mentioned if variables were tested for normality and homocedasticity.

- Considering Cronbach's alpha values above .70 as acceptable is odd to me, since Nunnally & Bernstein considers that for group comparison values should be above .80 and for individual interpretations values should be above .90. See Nunnally, J. C., & Bernstein, I. H. (1994). The Assessment of Reliability. Psychometric Theory, 3(1), 248-292.

Final comments:

- Please, note that where it reads DSM-V should read DSM-5. This is a mistake mentioned several times throughout the manuscript.

- Please review the language of the paper carefully to be certain that references to reliability, validity, and other psychometric terms conform to the recommendations in the 2014 AERA, APA, NCME Standards for Educational and Psychological Testing. For example, be sure when you discuss reliability that reliability refers to test scores, not tests, and that validity refers to the validity of test score interpretations—again, not to tests. Also when authors name “reliability was adequate” specify if it refers to internal consistency.

6. PLOS authors have the option to publish the peer review history of their article (what does this mean?). If published, this will include your full peer review and any attached files.

Reviewer #1: No

Reviewer #2: No

---

## [Author Response · Author response to Decision Letter 0]

7 Jun 2023

Review Comments to the Author

Reviewer #1: 

I thank you for the opportunity to read the article which deals with a very important area within alcohol research, namely validating measuring instruments in different cultures. The aim of this study was to validate the AUDIT and its subscales in Tanzania and also to investigate the predictive value of these. However I identified some deficiencies.

Answer: Thank you for your kind words and dedication towards revising our manuscript.

Question 1 - The sample was an availability sample of 259 people who were asked on the street to complete questionnaires, which means that it is difficult to generalize the results to others than those who were not asked.

Thank you for raising this point. We agree that the sample size is a limitation of study, as stated in the last paragraph of our Discussion. We have made slight modifications to the text to address the nature of our study as an evaluation of psychometric evidence rather than the validation of the instrument. 

Question 2 - On page 13, the authors define reliability and validity. The definition of validity is limited to conceptual validity. Reliability can also be defined as the relationship between different items in the scale.

Thank you for your suggestion. We agree that our definition was not comprehensive enough. To address this issue we decided to remove the definition from the introduction.

Question 3 - To diagnose alcohol use disorder, a self-rating scale with 11 items based on DSM-5 was used. However, there is no information on the validity of the scale. To quantify standard drinks, the American norm for standard drink has been used. However, it may be difficult for the participants in the study to translate the amount of alcohol they drank into standard American drinks.

Thank you for your question. Indeed, anticipating this challenge we trained local researchers to use culturally relevant estimates based on traditional alcohol percentages and drinking container amounts to estimate consumption. These estimates are currently in use for numerous projects. We have disclosed information about these estimations in the “Instrument” (Methods 2.4, end of first paragraph) and “limitations” (Discussion, last paragraph) sections.

Question 4 - The data has been analyzed with confirmatory factor analysis, which is a hypothesis testing method, but there is no information about which hypothesis is being tested. It should be clearer that it is a hypothesis about a one-dimensional model that is tested with a confirmatory factor analysis. Convergent validity means a relationship between two measuring instruments, but the authors have not stated which instrument was used in the calculation. The authors write that 127 participants had AUD divided into 50 mild, 30 moderate and 72 severe. Then the total will be 152 participants.

We are thankful for the points discussed. We have made changes to the objective (last paragraph of Introduction) to better reflect the nature of the hypothesis being tested. We have corrected the text to make it clearer that we did not calculate the convergent validity. Finally, we corrected the numbers on the first paragraph of the Results section.

Question 5 - In terms of sensitivity and specificity, the numbers in the text does not agree with what is stated in table 3.

Thank you for highlighting this error. We made corrections in the text to address it.

Reviewer #2:

 The study entitled "Clinical validation of the Alcohol Use Disorders Identification Test (AUDIT) in Moshi, Tanzania aim to validate the Alcohol Use Disorders Identification Test (AUDIT) in the general population of Moshi (Tanzania), and evaluate the usefulness of the tool at predicting alcohol-related harms.

The topic is interesting given the importance of alcohol-related health problems and the study is well conducted. However, there are important limitations that seems difficult to solve. Basically, the study is more a translation and adaptation than a validation. This is particularly so, because a convenience sample of 259 adults from Moshi (Northern Tanzania) con not be generalise to the whole country. The authors themselves correctly acknowledge this important limitation: “[the sample ]was not representative of the population”

The following commentaries should also be considered in order to improve the quality of the article for publication. I would like to emphasize that the study is very well conducted but with a very limited and not representative sample.

Answer: We are thankful for the kind comments and careful review of our work. We agree that the sample size is indeed a limitation of our study. We have made changes throughout the text to address the correct nature of our study as the evaluation of psychometric evidence, rather than a validation of the instrument. 

Methods

Question 1 - Excluding participants who do not drink alcohol is not a good idea because prevent the study to examine prevalence of alcohol and risk overestimating alcohol related disorders. They should have been included and categorised with 0.

Alcohol use is highly prevalent in our study setting. In our patient sample, only 3 participants reported never having consumed alcohol before. In this sense, we decided to remove them from the analysis, focusing on a subsample of participants that reported at least some degree of alcohol consumption.

Data analysis and results.

Question 2 - It is not mentioned if variables were tested for normality and homocedasticity.

Thank you for your question. We have applied an approximation to ordinal data for the internal structure calculations, and therefore we did not check for normality. Furthermore, we believe that normality does not apply because of the principle of tau-equivalence. 

Question 3 - Considering Cronbach's alpha values above .70 as acceptable is odd to me, since Nunnally & Bernstein considers that for group comparison values should be above .80 and for individual interpretations values should be above .90. See Nunnally, J. C., & Bernstein, I. H. (1994). The Assessment of Reliability. Psychometric Theory, 3(1), 248-292.

Thank you for the comment. Even though we had used a reference of Cronbach values above 0.70, all of our internal consistency values were above 0.80, as indicated by Nunnally and Bernstein (1994). We use it as a reference as recommended. 

Final comments:

Question 4 - Please, note that where it reads DSM-V should read DSM-5. This is a mistake mentioned several times throughout the manuscript.

It was corrected throughout the manuscript. Thank you for the careful revision.

Question 5 - Please review the language of the paper carefully to be certain that references to reliability, validity, and other psychometric terms conform to the recommendations in the 2014 AERA, APA, NCME Standards for Educational and Psychological Testing. For example, be sure when you discuss reliability that reliability refers to test scores, not tests, and that validity refers to the validity of test score interpretations—again, not to tests. Also when authors name “reliability was adequate” specify if it refers to internal consistency.

The Standards for Educational and Psychological Testing was used and the terms were revised throughout the manuscript. Thank you for the comment.

---

## [Editor Report · Decision Letter 1]

14 Jun 2023

Clinical Evaluation of the Alcohol Use Disorders Identification Test (AUDIT) in Moshi, Tanzania

PONE-D-22-15181R1

Dear Dr. Catherine A. Staton,

We’re pleased to inform you that your manuscript has been judged scientifically suitable for publication and will be formally accepted for publication once it meets all outstanding technical requirements.

Kind regards,

Sebsibe Tadesse, PhD

Academic Editor

PLOS ONE

---

## [Editor Report · Acceptance letter]

26 Jun 2023

PONE-D-22-15181R1 

Clinical Evaluation  of the Alcohol Use Disorders Identification Test (AUDIT) in Moshi, Tanzania 

Dear Dr. Staton:

I'm pleased to inform you that your manuscript has been deemed suitable for publication in PLOS ONE. Congratulations! Your manuscript is now with our production department. 

Kind regards, 

on behalf of

Dr. Sebsibe Tadesse 

Academic Editor

PLOS ONE